# Multi-Scale Application of Advanced ANN-MLP Model for Increasing the Large-Scale Improvement of Digital Data Visualisation Due to Anomalous Lithogenic and Anthropogenic Elements Distribution

Robert Šajn [1,*], Trajče Stafilov [2], Biljana Balabanova [3] and Jasminka Alijagić [4]

[1]  Geological Survey of Slovenia, 1000 Ljubljana, Slovenia
[2]  Institute of Chemistry, Faculty of Science, Ss. Cyril and Methodius University, 1000 Skopje, North Macedonia; trajcest@pmf.ukim.mk
[3]  Faculty of Agriculture, University "Goce Delčev", 2000 Štip, North Macedonia; biljana.balabanova@ugd.edu.mk
[4]  Independent Researcher, 1000 Ljubljana, Slovenia; jasminka.alijagic@gmail.com
[*] Correspondence: robert.sajn@geo-zs.si; Tel.: +386-1-2809-769

**Abstract:** The main objective of this paper is to compare and improve spatial distributions models for Pb and Cu in air and soil using the universal kriging and ANN-MLP at the macro regional scale. For this purpose, both models have been applied for visualization of a spatial distribution of lead (Pb) and copper (Cu) in a morphologically and geologically complex area. Two river basins in the eastern part of North Macedonia, have been selected as the main research region due to the extensive anthropogenic impact of long-lasting mining activities, with emphasis on the specific geochemistry of the area. Two environmental media (soil and moss) have been selected as they are much more available as space from biospheres submitted for destruction processes globally. Surface soil and moss as bio-indicator element measurements were submitted in correlation with geospatial data obtained from DEM, land cover data, and remote sensing, and are incorporated into spatial distribution mapping using an advanced prediction modeling technique, ANN-MPL. Both methods have been further compared and evaluated. The comparative data outputs have led to the general conclusion that ANN-MPL gives more realistic, reliable, and comprehensive results than the universal kriging method for the reconstruction of main distribution pathways. The more the factors influencing the process of distribution of the elements increase, the more the use of ANN-MPL improves.

**Keywords:** trace elements; soil; moss; spatial distribution; artificial neural network–multi-layer perceptron; universal kriging

## 1. Introduction

Long-term lithogenic degradation effectively disrupts a number of natural conditions in the environment. In addition, the amount of atmospheric emissions is increasing rapidly worldwide. Emissions from mines, mine activation, smelters, metal processing plants, fertilization, and combustion contribute to the anthropogenic inputs of heavy metals in the environment [1–4]. Surface lithological degradation has been significantly enhanced by long-term mining activities and processing. Mining and flotation waste continue to be a threatening source of degradation, even long after mining activities have stopped. The atmo-lithosphere area is a critical space of the biosphere, which directly affects the well-being of the human population. Well-developed countries implement continuous monitoring programs in order to identify and anticipate anomalous areas and occurrences that negatively affect the health of the population and contribute to ecosystem destruction.

In the late 1960s, Scandinavian countries introduced suitable monitoring models for atmospheric deposition of heavy metals and other trace elements [4]. Since then, many

countries have used bryophytes in national and multinational air pollution studies to obtain valuable information about spatial and temporal changes in trace metal deposition [5–7]. Attributable to the effect of potentially toxic elements on living organisms, information on the amount and distribution of heavy metals in the environment is of great importance. Consequently, many tools have been developed and tested to help researchers to analyze the processes of the distribution of contaminated particles and the level of contamination in air and soil. Focusing on digital soil mapping (DSM), the variability of a target soil property is explained by its relationship with soil-forming factors, including topography, land use, climate, vegetation, and soil type [1,8]. DSM methods provide accurate maps of soil properties, easy access to various types of high-quality data, an increase in the speed of computing processes, and they are less time consuming and expensive. The high capability of machine learning methods using available environmental data to predict trace metals in soil at a large scale has been successfully proved [9]. Multisource geospatial data can be used for the mapping of spatial patterns of trace elements in urban topsoil [10], or of some soil parameters, such as pH [11] and soil organic carbon (SOC) [12], or in updating a conventional map [13]. Furthermore, numerous statistical techniques have been applied to present levels of contamination, such as multiple linear regression [14], kriging [15], and regression kriging [16]. In the last few years, a few studies have been applied from the machine learning field, such as artificial neural networks [17], boosted regression trees [18], and random forests [19]. Those machine learning methods overcome the imperfections in parametric and non-parametric statistical methods, including overfitting, autocorrelation, and non-linearity [20], and, in that way, yield more accurate spatial models.

The present investigation selected two models for the comparison of the realistic output of air distributed pollutants. The ANN model can be used to predict air pollutants in areas where monitoring sampling points are set up. Air quality has to be generalized for a whole area where there are no monitoring sampling points. Interpolation can be applied to the measured data and can be used to obtain the air quality at unmonitored locations. Kriging is a geo-statistical interpolation method that is flexible in terms of input and output data and takes the location into consideration when predicting the values of desired parameters. Kriging predicts the values of parameters along with prediction errors and probabilities. The success of kriging is indicated by prediction errors and cross validation. This paper proposes an integrated model combining and comparing the techniques of ANN and a kriging model, and examines the results obtained by using both models separately to predict the levels of affected areas with higher contents of Pb and Cu in air distributed dust.

Therefore, the presented investigated study area considers the Bregalnica River Basin and Kriva Reka Basin, located in the eastern part of the North Macedonia, where the number of variables that influence the accuracy of the applied methods is larger compared to our previous study, performed in Vareš, Bosnia, and Herzegovina [21]. For more than ten years, intensive monitoring has been carried out in both river basins for the distribution of chemical elements in lithological media and in terms of the emission distribution of contaminated particles in air [22–27]. The specificity of the researched area consists in the characteristic geochemical natural distribution and anthropogenic anomalies of long-term mining activities. In addition, this area has the natural phenomenon of a compilation of old and new volcanism, which provides unique conditions of distribution along the alluvial plains of both river basins. The Bregalnica River Basin is naturally enriched with Pb, Zn, Cu, and Cd due to its lithological and geological characteristics. Although these elements represent background enrichment, long-term exploitation has resulted in elevated levels of these elements, well above the natural background. Fine particles from mining and flotation wastes originating from nearby mines (Bučim Cu Mine, Sasa and Zletovo Pb-Zn Mine) have been dispersed by winds at smaller and larger distances from the emission source, posing a hazard to living organisms. The pollution in this area has, therefore, been the main subject of many investigations [22–28]. In the northern direction of the Bregalnica River Basin is the Kriva Reka Basin, an area in which similar conditions of litho-geological phenomena are found. For a period of 10 years, these areas have been monitored as separate research

regions. Preliminary studies in this research area have shown that characteristic geological phenomena occur on a similar basis, as do the conditions of anthropogenic exploitation of mineral resources.

Thus far, conducted research and mapping for these areas have experienced a number of anomalies, in terms of insufficiently precise interpolation of visualized data for which researchers cannot provide a logical explanation, supported by the processes of anthropogenic and natural influences. Despite the uniqueness of the mentioned region, globally, there are many affected areas, not only in terms of the distribution of heavy and toxic metals, but also in terms of other xenobiotics present in the environment. Thus, the main objective of this paper is to present the results of a systematic study of the spatial distribution of contamination of different elements in soil and mosses, using linear and nonlinear mathematical methods, combining chemical analyses and different geospatial parameters. The main purposes of this study were:

(i)   Generating soil–air model monitoring in conjunction with linear and nonlinear modeling techniques, using universal kriging and artificial neural network multilayer perceptron (ANN-MLP);

(ii)  To compare the two modelling techniques;

(iii) To identify the main distribution pathways and element distributions as a function of distance from the contamination source;

(iv)  To evaluate the realistic size of the affected area;

(v)   To evaluate the efficacy of digital soil mapping and ANN-MPL.

Based on the set target of the multi-methodological approaches, we believe that this research will provide a wide range of available opportunities for: (a) the design and development of multi-aspects monitoring systems; (b) describing methods and procedures for pollution risk assessment; (c) developing the synthesis of monitoring data with mapping and distribution, indicating a realistic presentation of affected areas with anomalous phenomena. The testing range includes almost 40 elements. However, this study summarizes the expressive data of the distribution of Pb and Cu in the region of both the Bregalnica River Basin and Kriva Reka Basin, North Macedonia.

## 2. Materials and Methods

### 2.1. A Geographic Description of the Study Area

The investigation subarea of the Bregalnica River Basin covers an area of about 4300 km$^2$ and 1110 km$^2$ of the Kriva Reka Basin. In total, an area of almost 5400 square kilometers was included in this research (Figure 1). The Bregalnica River Basin occupies two valleys—the Maleševska and the Kočani Valleys. The Kočani Valley represents the middle course of the Bregalnica River and includes a significantly explored agriculturally area [29]. In general, the region is characterized by a moderate continental climate [29]. The average annual temperature is about 13 °C [29]. The most frequent winds are from the west, with a frequency of 199‰ and a speed of 2.7 m/s; the east winds have a speed of 2.0 m/s, with a frequency of 124‰, and have an important influence on the distribution of atmospheric dust [30]. This area is characterized by an average annual precipitation of about 500 mm and is regulated by Mediterranean cyclones. The subtropical anticyclone results in warm and dry summers, while the central area of the region is considered the driest. From there, due to the increasing influence of the Mediterranean climate or the increase in altitude, average annual precipitation increases in all directions [29]. The area of Bregalnica River Basin is characterized by several significant emission sources of potentially toxic metals, including the copper mine and flotation "Bučim" near Radoviš, the lead and zinc mines "Sasa" in the vicinity of Makedonska Kamenica and "Zletovo" near Probištip. Flotation of copper minerals produces about 3,950,000 tons of flotation residues annually [31], while at the Sasa mine this number is between 70,000 and 100,000 m$^3$ [32,33]. In the flotation tailings of the Zletovo mine, it is estimated that the total amount of deposited dump is about 14 Mt [34]. The tailings are exposed to airflow, which leads to the dispersion of contaminated particles. In addition to wind, which may disperse and deposit particles

over short or long distances, ore residues may be washed out by rainwater, resulting in the leaching of the metals present and their entry into soils, rivers, and underground waters [27].

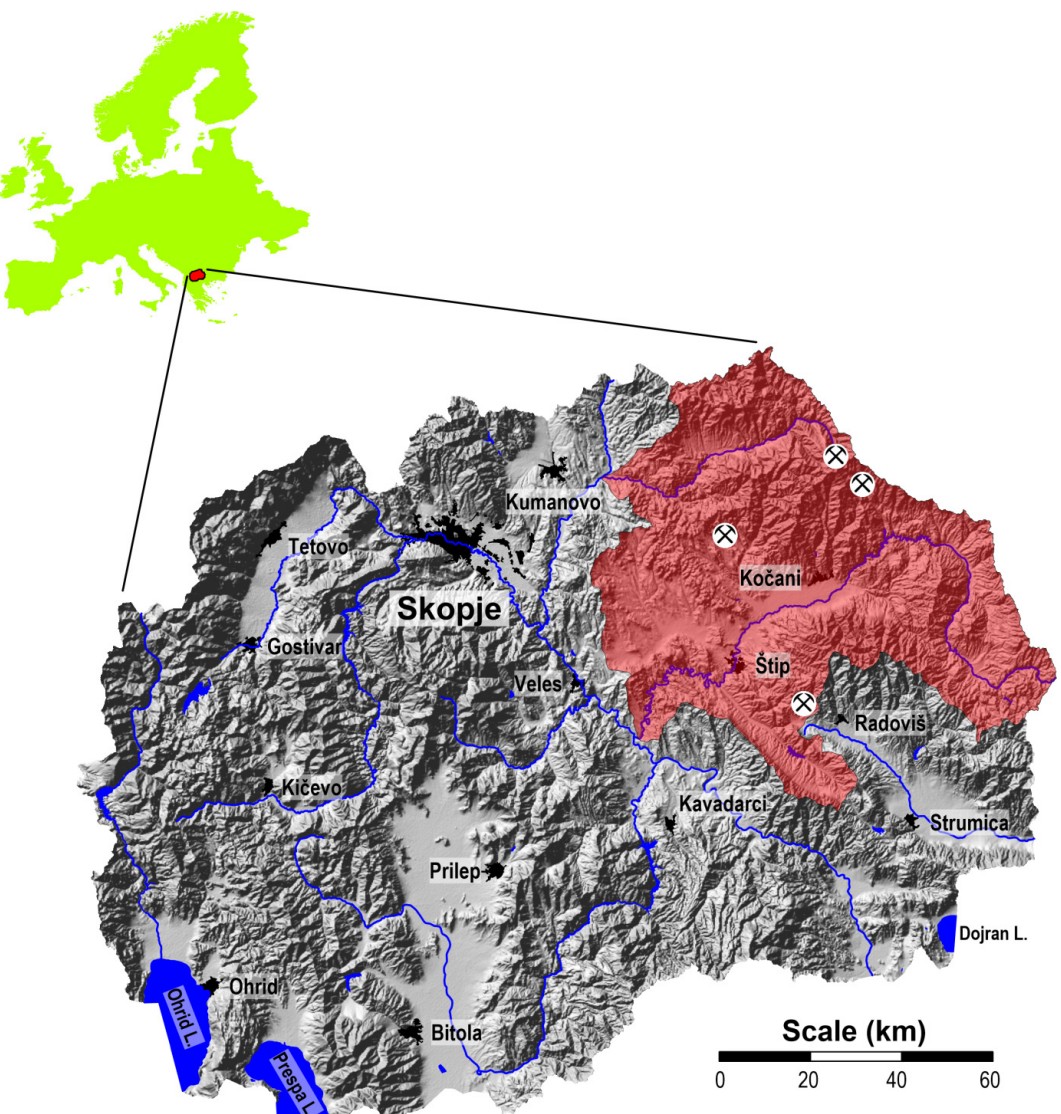

**Figure 1.** Location of the study area.

The region of the Kriva Reka Basin is characterized by a moderate continental climate. The altitude varies between 600 and 1500 m. The most frequent winds in the region are those from the west with a frequency of 199‰ and speed of 2.7 m s$^{-1}$; winds from the east have a frequency of 124‰ and speed of 2.0 m s$^{-1}$ [29]. Climatic condition in the region allows for the air-distribution of fine dust particles, generated as a result of exposure flotation tailings in the open [29]. One of the potential emission sources of hazardous heavy metals in air in the south-eastern part of North Macedonia is the Toranica lead-zinc mine, in the vicinity of the town of Kriva Palanka. This Pb-Zn mineral deposit has been exploited since 1987. Pb-Zn production at the mine occurred between 1987 and 2000, reaching a total amount of ~3 × 106 tons of excavated ore with an average content of 6.5% Pb-Zn [32]. Milling and flotation occurred at the mine and there is a tailings dam below the mine site with a culvert directing the Toranica River beneath the dam. The flotation plant for the production of Pb and Zn concentrates is situated close to the mine. Flotation tailing disposal leads to uncontrolled introduction of higher amounts if dust that contain higher contents of Pb and Zn.

### 2.2. Sampling, Sample Preparation, and Chemical Analyses

A systematic sampling, using a regular sampling grid for both sampling media, the moss and soil, was applied. The sampling grid for both sampling materials was denser around the mining areas. The entire covered area consisted of 409 samples of soil and 286 moss samples. Regarding soil samples, out of 409 sites, 218 samples were collected from a regular sampling grid of 5 × 5 m; the 126 sampling sites were mostly collected from around the mining areas, and 65 soil samples were collected from alluvial soils. The soil samples were collected from a depth of 0–5 cm (Figure 2). To obtain a representative composite sample, five subsamples were collected at each site in a radius of 50−100 m. The total number of moss samples was 286 (Figure 2). The 195 sampling sites corresponded to the sampling sites collected in a regular grid of 5 × 5 m, while the additional 91 sampling sites were mainly located around mines. To avoid the problem of association with the predicted sampling site due to the presence or absence of selected species, samples of different moss species were taken from the same slope line (Homalotecium lutescens, Hypnum cupressiforme, Pleurozium schreberi and Hylocomium splendens). The interchangeable uses of various species have been demonstrated for the mentioned moss species and were given by Balabanova et al. [31].

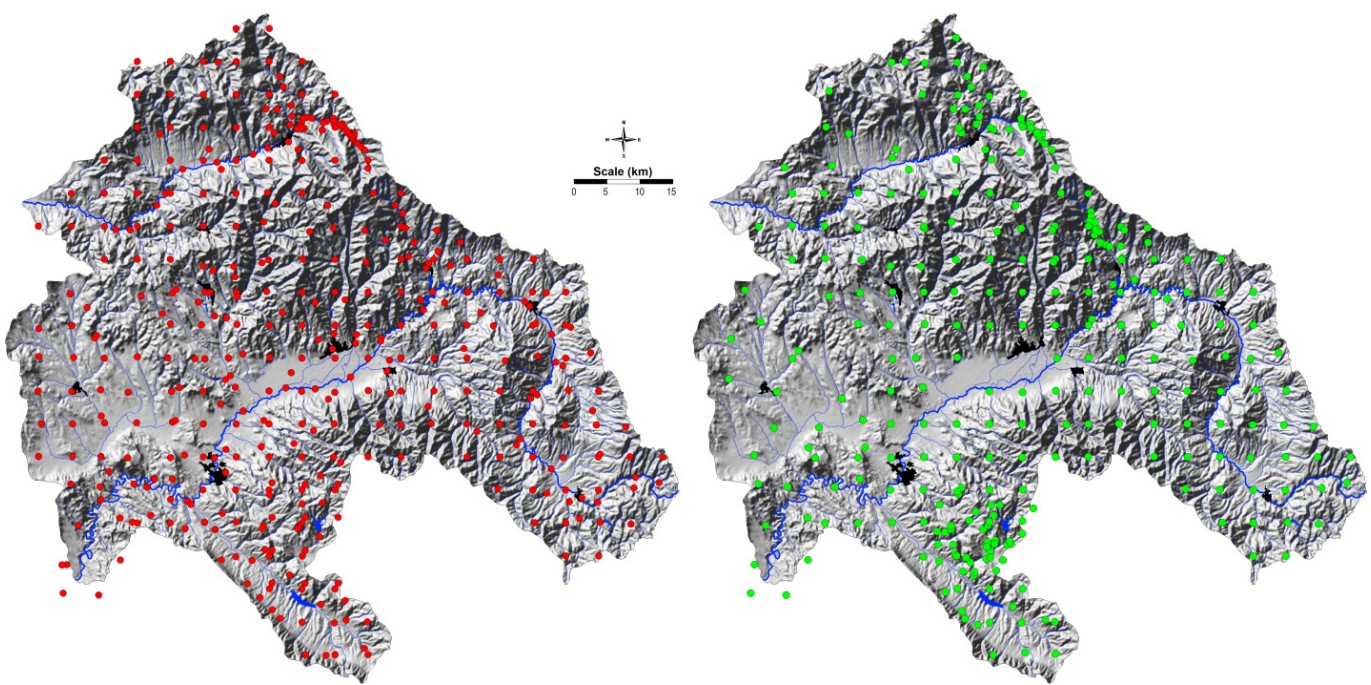

**Figure 2.** Map of soil (red points) and moss sampling sites (green points).

Pre-treatment of the soil samples included cleaning and drying to an absolute amount of mass. As a part of the standard soil sample preparation procedure, diverse combinations of sieving, milling, and grinding of samples was performed in order to reach final particles of 125 μm, before analytical treatment. The samples were chemically prepared using wet digestion, applying a mixture of acids in accordance with international standard ISO 14869-1:2001. For the measurements of the elemental concentrations in soil digests, inductively coupled plasma with mass spectrometry (ICP-MS) was performed. The instrumental conditions of the mass spectrometer were published by Balabanova et al. [23–28,31].

Moss sample collection was conducted in accordance with the adopted protocol, published by Harmens et al. [5]. Closed digestion using a microwave digestion system was used for moss sample preparation (CEM, Mars model). The procedure of moss sample digestion was given by Balabanova et al. [31]. Determination of Pb and Cu concentrations in moss samples was conducted using ICP-MS, and the whole procedure was given by Balabanova et al. [31].

Both analytical instruments were used to obtain a higher accuracy for the element contents in the samples. For both instrumental analysis methods, validation was applied in accordance with standard QA/QC criteria.

### 2.3. Environmental Covariates

ANN-MLP was applied as a prediction method in order to prepare a high quality spatial distribution map, and to increase efficacy and accuracy. Due to the specificity of the terrain, where the climatic conditions and the relief have an intensive influence on each other, several aspects of dependence were included, in addition to the geological and lithological characteristics of the area. For this purpose, we selected 23 environmental covariates that were used in the training data to better describe and understand the relationships of environment–soil and environment–moss. Thus, we included the following parameters: three categorical inputs—geology units (Quaternary alluvium, Quaternary terrace, Neogene clastites, Neogene pyroclastites/vulcanites, Paleogene flysch, Paleogene volcanic sedimentary rocks, Paleozoic shales, Rifeous shales, Proterozoic shales/gneisse, Proterozoic plutonites); Quaternary divison (Bregalnica-upper flow, Bregalnica-lower flow, Kriva Reka, Kamenica, Zletovska Reka, Lakavica, Svetinikolska Reka); and land use (forest, open space, agriculture area, urban area). Seventeen continuous inputs were used: X coordinate, Y coordinate, distance from Sasa/Toranica Pb-Zn mine, distance from Zletovo Pb-Zn mine, distance from Bucim Cu open pit, altitude (m), slope, plan curvature, profile curvature, tangent curvature, and Landsat spectral bands: 0.45–0.52 μm (blue), 0.52–0.61 μm (green), 0.63–0.69 μm (red), 0.76–0.90 μm (IR), 1.55–1.75 μm (IR), 2.08–2.35 μm (IR), and 10.4–12.5 μm (thermal).

For the modeling process, a recall grid was used, which meant that the entire study area was fractionated into 100 × 100 m grid cells. The total number was 540.497. Our goal was to describe each recall point with as many environmental covariates as possible and to prepare a realistic, reliable, and comprehensive spatial distribution map that simultaneously reconstructed the main distribution pathways. Furthermore, through the enrichment factors and ratios among some elements associated with the geological data, it was possible to identify geochemical changes in the environment.

### 2.3.1. Geology

Georeferenced geological maps were then attached to the study area (Figure 3). The main lithological units of the study area were isolated and used in the training dataset [34]. The studied area lies on two main tectonic units—the Serbian–Macedonian Massif and the Vardar Zone [35]—and is characterized by several subvolcanic areas. The polyphasal Neogene deformations play an important role in the development of this area. These deformations, associated with volcanic activity, influence the gradual formation of reefs, as well as the formation of deposits in existing basins, and are associated with alternating periods of fast and slow landslides, which are accompanied by variable deposition from the middle Miocene to the end of the Pleistocene. Cenozoic volcanism represents a more recent extension in the Serbian–Macedonian massive and the Vardar zone. The oldest volcanic rocks occur are the areas of Bučim, Damjan, the Borov Dol district, and in the zone of Toranica, Sasa, Delčevo and Pehčevo [28,32]. These older volcanic rocks were formed in the mid-Miocene from sedimentary rocks that represent the upper age limit of the rocks. The origin of the oldest volcanic rocks is related to the Oligocene—the early-Miocene period.

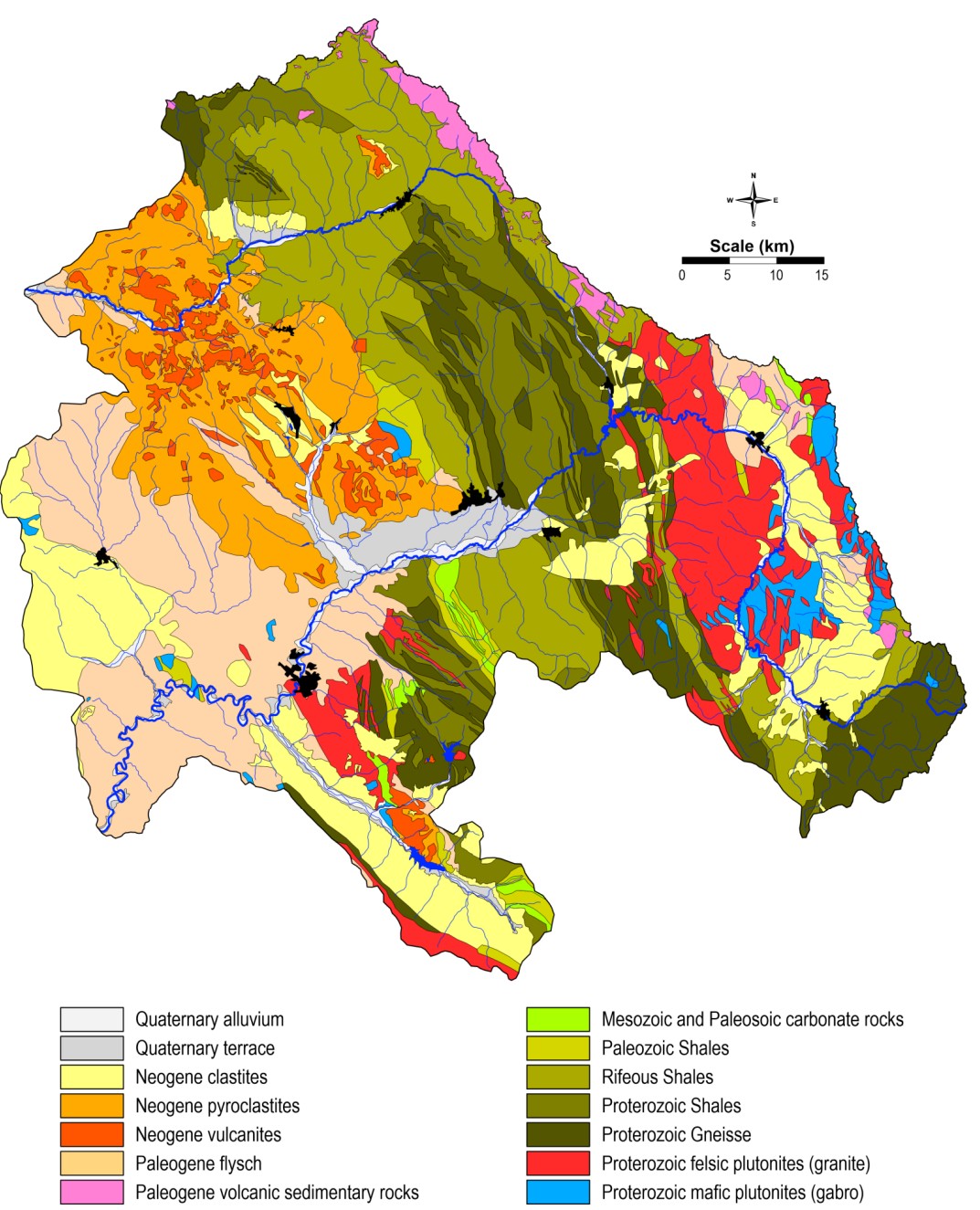

| | |
|---|---|
| Quaternary alluvium | Mesozoic and Paleosoic carbonate rocks |
| Quaternary terrace | Paleozoic Shales |
| Neogene clastites | Rifeous Shales |
| Neogene pyroclastites | Proterozoic Shales |
| Neogene vulcanites | Proterozoic Gneisse |
| Paleogene flysch | Proterozoic felsic plutonites (granite) |
| Paleogene volcanic sedimentary rocks | Proterozoic mafic plutonites (gabro) |

**Figure 3.** Basic lithological units.

Volcanic rocks are categorized as andesite, latite, quartz-latite, and dacite. Volcanism appears sequentially and, in several phases, forming sub-volcanic areas. On the other hand, pyroclastites are most frequently found in the Kratovo-Zletovo volcanic area, where dacites and andesites are the oldest formations [28,32].

Magmatic–hydrothermal fluids play an important role in formation of ore deposits in the study area. The mineralization of the Bucim Cu porphyric ore deposit is related to tertiary sub-volcanic intrusions of andesite and latite in a host of pre-Cambrian gneisses and amphibolites. The geology of the Toranica ore deposit comprises various rocks of metamorphic origin and igneous rocks of the Tertiary age. The mineralization of Zletovo is related to Tertiary calc-alkaline magmatic rocks (dacites and andesites) and it is found in a dacite volcano/sedimentary suite that has been altered to clay and micas. The main minerals are galena and sphalerite, but tetrahedrite, pyrothite, and chalcopyrite are also common. Sasa mine mineralization is localized along the contacts between the Miocene calc-

alkaline igneous body (latites and dacites) and graphite-chlorite-sericite schists, gneisses, and limestones [36,37].

### 2.3.2. Land Cover

The CORINE land cover (CLC) dataset was obtained from the Copernicus Global Land Service (CGLS) (https://land.copernicus.eu/global/, accessed on 25 May 2018). CLC databases have the following important characteristics: (a) they are developed regularly; (b) support a wide range spatial analysis; (c) enable identification of various land-use types; and (d) support forecasting. For this study, four main units were isolated and used in the machine learning process (Figure 4). Due to fact that CLC is a less reliable tool for analyses conducted on a large scale, it was necessary to make corrections in some places.

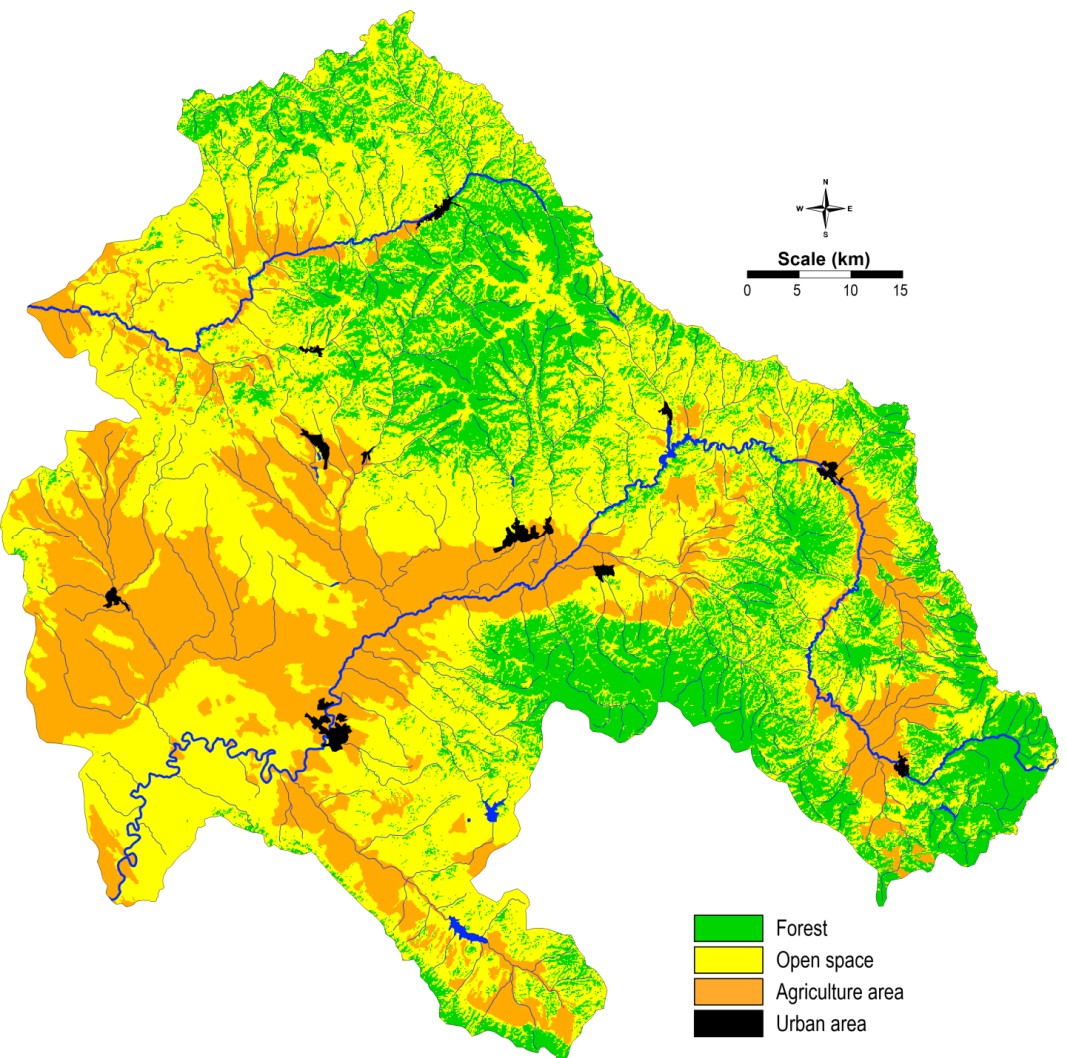

**Figure 4.** Land cover map.

Regarding the demographic structure in the region, 65% of the populated areas belong to the category of urban areas, while 35% of the total populated areas are categorized as rural areas. In the research region, there is an unequal distribution of urban and rural populations. Considering land cover, the region is considerably diverse. Along the whole courses of the Bregalnica and Kriva Reka rivers, agricultural cultivated lands dominate. Pastures are also considered as agricultural lands, and are represented in yellow, as open areas. About 30% of the study area belongs to the forest regions, localized around the Maleševska Valley, represented by the Maleševski Mountains, and around the Kočani Valley,

represented on one side by the Osogovo Mountains and by Plačkovica on the other. Kočani Valley is one of the most used agricultures lands along the Bregalnica River Basin. Along the upper course of Kriva Reka brown forest soil, as well as leptosol and regosol dominate. The rest of the area is characterized by the occurrence of complex of humic calcaric regosol and smolnica and rendzina as the dominant pedological units. Quite complex pedological units composites are found in the Bregalnica River Basin (http://www.maksoil.ukim.mk/masis, accessed on 25 May 2018). Most intensive agricultural land use occurs in Kočani Valley, which occupies the middle course of the river. The main pedological units are complexes of vertisol, regosol and leptosol.

### 2.3.3. Digital Elevation Model (DEM)

A DEM (Figure 5) was downloaded from the Shuttle Radar Topographic Mission (SRTM) with a 30 m spatial resolution and was used for the extraction and preparation of a set of topographic attributes, such as altitude, terrain slope, aspect, plan terrain curvature, profile terrain curvature, tangent terrain curvature, and wind effect. ANN multilayer perceptron was employed to model infiltration using data derived from these covariates (Figure 6). Training parameters, such as the stopping criteria and the type of transfer function, affected the efficiency and the generalization capability of the ANN. The ANN resulted in a satisfactory network. Network performance was also found to be related to the input, as the accuracy of the ANN model was higher for soil types with higher proportions in the input data.

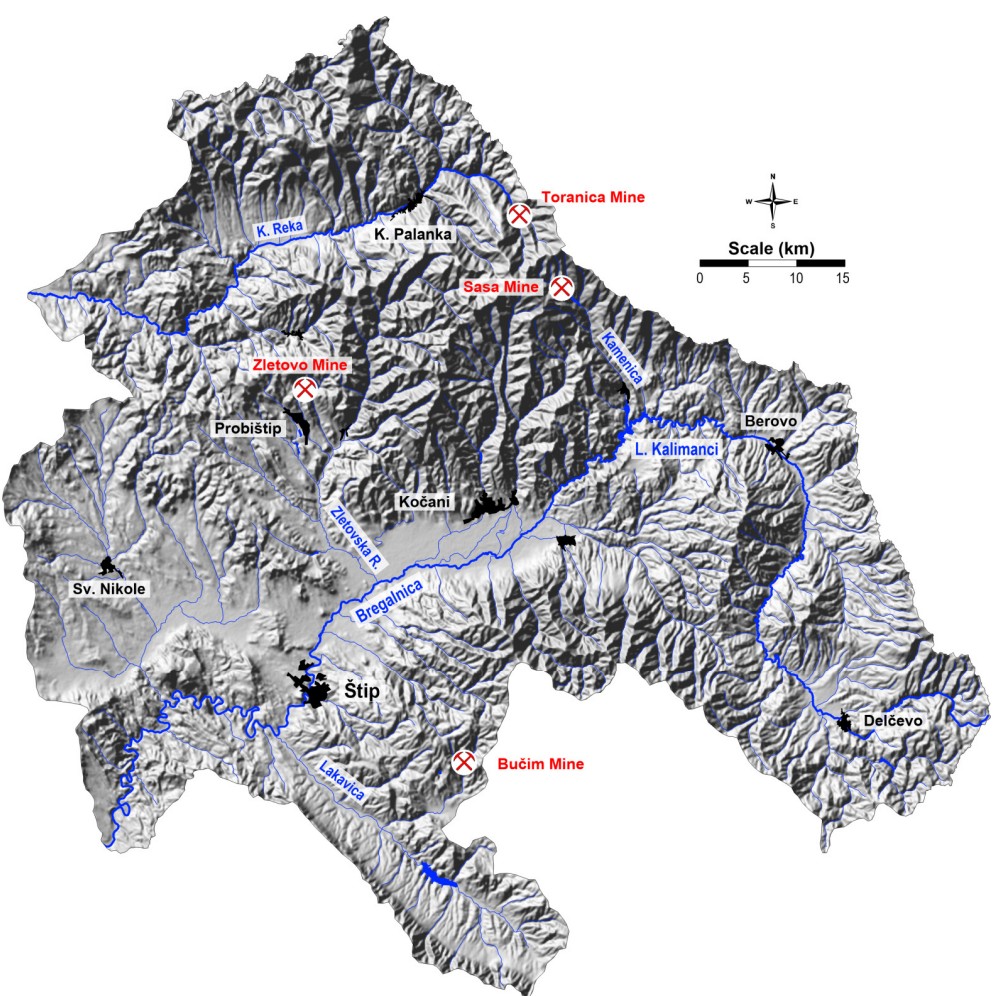

**Figure 5.** Digital elevation model of the study area.

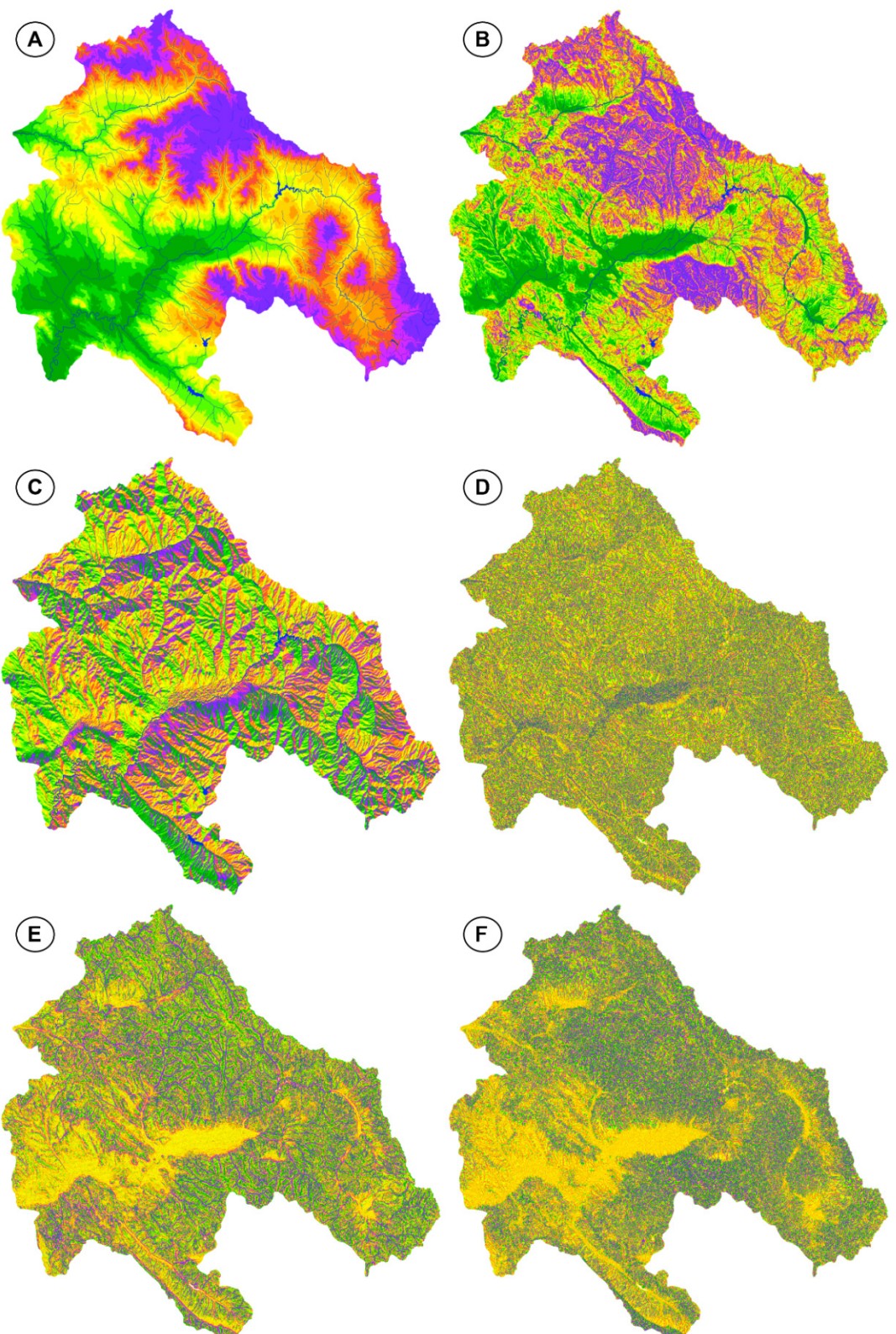

**Figure 6.** Topographic attributes ((**A**) altitude above sea level (absolute), (**B**) terrain slope, (**C**) aspect (insolation), (**D**) plan terrain curvature, (**E**) profile terrain curvature, (**F**) tangent terrain curvature).

2.3.4. Distances from the Main Sources of Contamination

Due to fact that the concentrations of trace elements are strongly correlated with distance and depend highly on the source of contamination, we decided to include these

parameters as well. For this purpose, three main distances from the mines Sasa—Toranica, Zletovo and Bučim—were calculated and are presented in Figure 7.

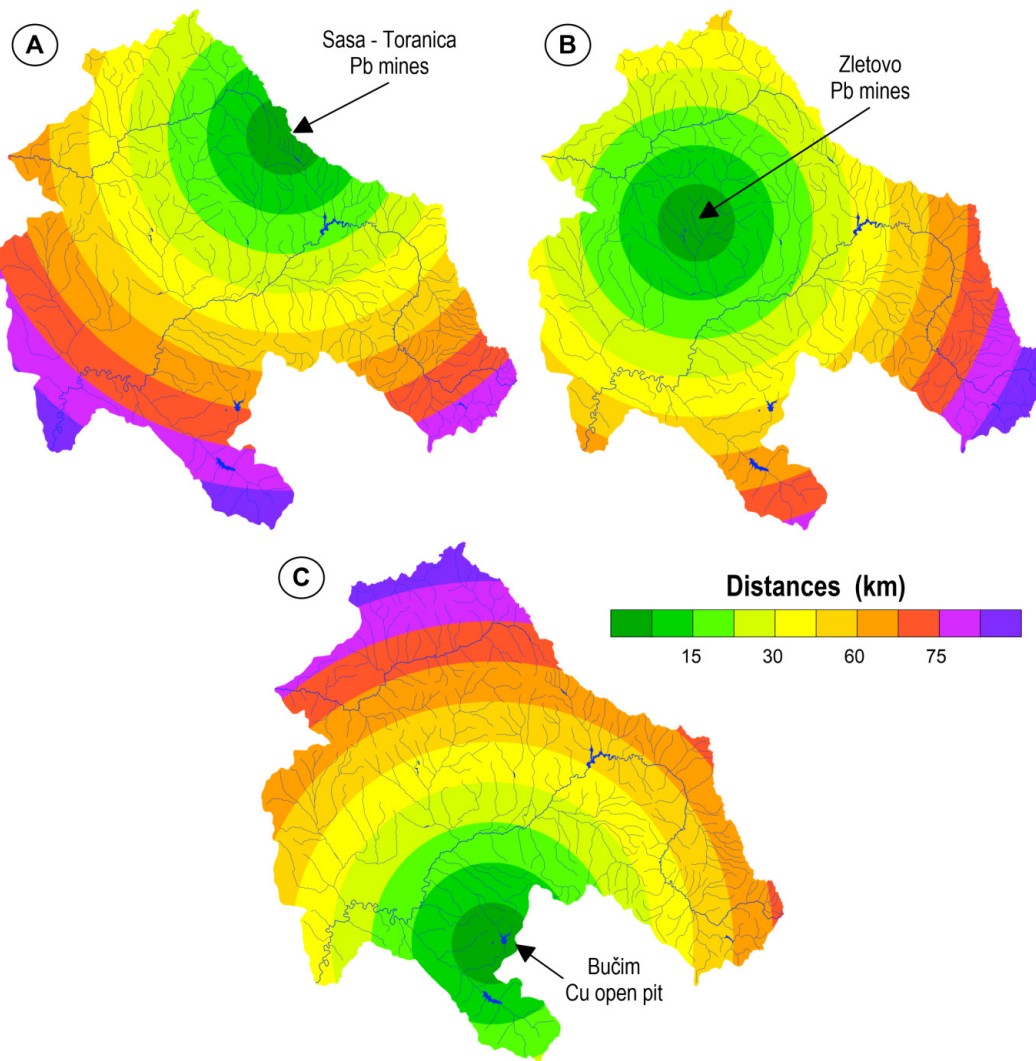

**Figure 7.** Distances from the main sources of contamination. (**A**) Sasa–Toranica Pb mine, (**B**) Zletovo Pb mines, (**C**) Bučim Cu open pit.

### 2.3.5. Landsat Images

Landsat image with a spatial resolution of 30m were downloaded from https://earthexplorer.usgs.gov/ (accessed on 25 May 2018). The images were geometrically and radiometrically corrected (Figure 8). Information from the Landsat multispectral images was extracted and incorporated into the training dataset, which was used for the learning process.

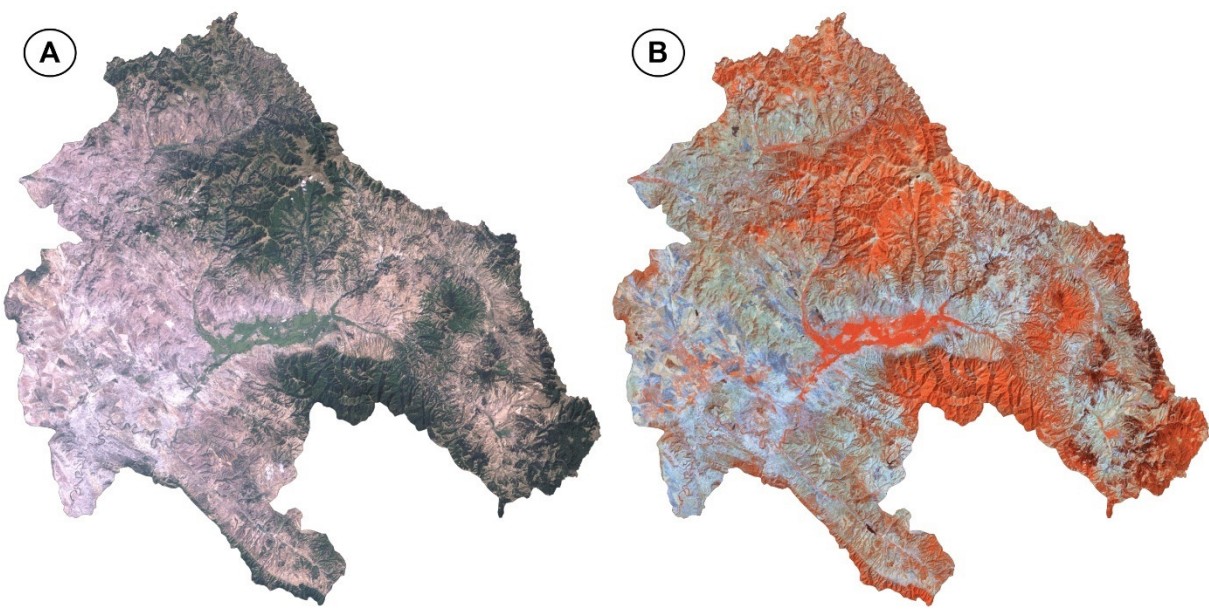

**Figure 8.** Landsat spectral bands. (**A**) Join visible spectrum, 0.45–0.69 μm (B10-B20-B30), (**B**) join Infrared spectrum, 0.76–2.35 μm (B40-B50-B70).

### 2.4. Data Transformation

Data variables have different ranges of values and units. To overcome these, data were standardized in order to provide equal attention during the extraction process and remove the effects of similarities between objects. In order to reduce the difference between extreme values, data transformation was performed. Quite often, is has been observed that environmental variables are Log-normal [38–40] or positively skewed [41,42] and consequently it was necessary to perform data transformations. Based on the results and our experience, we applied a Box-Cox transformation, which represents one of the most frequently used transformations in environmental sciences and geosciences [43].

The Box-Cox transformation is given by:

$$y = \frac{X^\lambda - 1}{\lambda}; \ \lambda \neq 0$$
$$y = \ln(\lambda); \ \lambda = 0$$

where $y$ is the transformed value, and $x$ is the value to be transformed. For a dataset $(x_1, x_2 \ldots x_n)$, parameter $\lambda$ is estimated based on the assumption that the transformed values $(y_1, y_2 \ldots y_n)$ are normally distributed. When $\lambda = 0$, the transformation becomes a logarithmic transformation.

### 2.5. Universal Kriging

Kriging is based on statistical models that include autocorrection—the statistical relationship between the measured points. Kriging is the most advanced interpolation method that accounts for both the distance dependence and direction dependence. This statistical geographic method in soil evaluation produces a predicted area of heavy metal distribution, considering certainty and prediction accuracy [38,39,44,45]. The dependence between the points predicting the spatial or geographical distribution is given by the following equation:

$$Z(S_0) = \sum_{i=1}^{N} \lambda_i Z(S_i)$$

where $Z(S_i)$ represents the value measured at location $i$; $\lambda_i$ means an unknown weighting of the measured value in location $i$; $S_0$ is the location of the prediction; and $N$ is the number of measured values.

This procedure is often used to estimate unsampled areas. It can also be used to build probabilistic models to prove unknown but estimated predicted values [46]. Maps produced using this method provide a visual representation of spatial variability in study areas. It can be used to represent soil properties in areas with potential natural hazards [47].

The geostatistical approach is divided into two parts: the calculation of an experimental variogram from the data and model fitting, the second part is estimation prediction at non-sampled locations. A variogram is used to measure the spatial variability of a regionalized variable, providing the input parameters for the spatial interpolation of the variogram kriging [47]. It can be taken with the following equation:

$$\gamma(x) = \frac{1}{2N(h)} \sum_{i=1}^{N(h)} [(Zx_i) - Z(x_i + h)]$$

where $y(x)$ means the semivariance at a given distance $h$; $Z(x_i)$ represents the value of the variable $Z$ at the $x_i$ location, and $N(h)$ means the number of pairs of sample points that are separated by the lag distance $h$ [17].

### 2.6. Artificial Neural Network–Multilayer Perceptron

ANNs represent a method of artificial intelligence based on an analogy of the human brain [48]. Artificial neural networks simulate biological neural networks, where the function of a neural network is determined by the model of highly interconnected neurons, the network structure, and the learning algorithm. The method ANN can be used to solve the specific problems-pattern recognition and data classification [9,10,12,17].

The multilayer perceptron (MLP) is the most frequently used ANN, especially in environmental studies. This method can be used to perform tasks such as feature matching and pattern recognition problems. Moreover, MLP can be used to prove the classification of linear inseparable patterns. MLP represent feedforward neural networks (FNNs) with multiple layers of units between the input and output layers [17,21]. The output of a neuron can be expressed by:

$$\xi = \sum_{i=1}^{n} w_i x_i - b = w^T x - b$$
$$y = \sigma(\xi)$$

where $x_i$ is the number of $i$th input, $w_i$ is the link weight from the $i$th input, $w = (w_i \ldots w_n)^T$, $x = (x_1 \ldots x_n)^T$, $b$ means a threshold or bias, meanwhile the $n$ is the number of inputs. The task of activation function $\sigma(\xi)$ is usually some continuous or discontinuous function to transfer the real numbers into the interval [42]. Aa activation function it can be also used the sigmoidal activation function. It can be given with the following form [49]:

$$\sigma(\xi) = \frac{1}{1 + e^{-\xi}}$$

Compared to other commercial and freely available software packages that allow users to implement neural networks relatively easily by offering only a limited number of training algorithms, the MLP offers an unlimited number of algorithms to achieve better results [50].

## 3. Results

### 3.1. Data Processing

Geospatial data, including DEM and land use maps, were further used in the modelling process and are a crucial part of creating a spatial distribution. These geochemical distribution maps as a final product are important to identify and understand the significance of the main geochemical processes throughout the area, to compile and document the geochemical background, and to distinguish between the natural and man-made influence on the element distribution.

The investigated region is selected as a test area for comparison between two spatial distribution techniques due to presence of natural enrichment as a consequence of various geological/lithological settings, complex geomorphology and combination of anthropogenic activities. The main focus of this paper is on presenting a distribution of Lead (Pb) and Copper (Cu), two elements that mostly characterise the anthropogenic impact at the study area. Beside these two elements, the contamination and distribution anomalies occur within other elements. The main purpose of this research work is not a presentation of total contamination but rather the modelling achievements in advanced computer science and applicability of the methodology. Advanced modelling technique, beside the analytical measurements, include much more attributes that significantly improved the final products, the spatial distribution maps.

Model stability and model signification had been tested for two selected elements, Pb and Cu which represents a dominant anthropogenic pattern. Basic statistical parameters between normal (raw data) and transformed data for both selected elements are presented in Table 1.

**Table 1.** Comparison between the statistical parameters of raw data (n = 409 soil, 286 moss) according to transformation.

| Element | Transformation | Material | X | Md | Min | Max | A | E | KS | $\chi^2$ |
|---------|----------------|----------|-----|-----|------|--------|-------|-------|------|------|
| Cu | Raw Values | Soil | 50 | 26 | 5.8 | 1600 | 8.8 | 100 | 0.33 | 590 |
| Cu | Raw Values | Moss | 11 | 5.9 | 1.2 | 300 | 8.3 | 83 | 0.33 | 520 |
| Pb | Raw Values | Soil | 390 | 32 | 5.2 | 16,000 | 6.6 | 50 | 0.40 | 1000 |
| Pb | Raw Values | Moss | 23 | 5.8 | 0.25 | 710 | 6.4 | 47 | 0.38 | 650 |
| Cu | Box-Cox | Soil | 1.7 | 1.7 | 1.2 | 2.1 | −0.06 | 1.1 | 0.04 | 30 |
| Cu | Box-Cox | Moss | 1.1 | 1.1 | 0.20 | 1.7 | −0.17 | 2.3 | 0.05 | 17 |
| Pb | Box-Cox | Soil | 1.9 | 1.9 | 1.2 | 2.4 | 0.14 | −0.18 | 0.05 | 22 |
| Pb | Box-Cox | Moss | 1.5 | 1.4 | −1.6 | 3.3 | −0.12 | 1.9 | 0.06 | 24 |

X—mean; Md—median; Min—minimum; Max—maximum; A—skewness; E—kurtosis; KS—Kolmogorov-Smirnov test; $\chi^2$—Chi-Square test.

Compositional data should be transformed since the raw data doesn't bring good results. Suitability of using the Box-Cox transformation is reflected in the fact that the distributions of natural values are significantly asymmetric. It could be observed, the mean and median values after transformation are lower. This anomaly could be partially explained by the fact that sampling grid is denser around the mining areas with dominant anthropogenic impact. In order to test the normality with skewness, kurtosis, Kolmogorov–Smirnov test, and Chi-Square test between raw data and Box-Cox transformation have been performed.

Obtaining an optimal neural network is usually one aim during the data processing. Because the MLP is prone to overfitting some form of regularization is usually performed for faster learning and generalization improvement. Because of advances in computational power, it may seem unnecessary to seek for efficient algorithms. However, in the pursuit of faster solutions, it is preferable that better and more stable solutions are found as well.

Modelling by ANN-MLP have been performed by using a large number of input data, 240 hidden units and 25 training networks. Modifications of input data, a number of neurons, as well as the number of training networks had been performed in order to obtain the best models. Based on the experience, more neurons and more architecture will acquire better results. The Neural network training architecture can be summarized as follows: 39 inputs, 250 hidden units and one output from this architecture we have 25 networks to train and 5 networks to retain. From the retain network, the average value had been calculated and furthermore used for mapping. During the network training data, the random sampling size can be summarized as follows: train 70%, test 15%, and validation 15%.

Each particular model is trained for 25 networks but only five networks had been used for further calculation. In addition, each training model contains a summary table (Table 2)

with following parameters: training perfection, test perfection, validation perfection, all perfection, training error, test error, validation error, training algorithm, hidden activation, and output activation.

**Table 2.** The training model output data.

| Element | Material | Training Perf. | Test Perf. | Validation Perf. | Training Error | Test Error | Validation Error |
|---------|----------|----------------|------------|------------------|----------------|------------|------------------|
| Cu | Soil | 0.80 | 0.45 | 0.58 | 0.008 | 0.016 | 0.018 |
| Cu | Moss | 0.70 | 0.53 | 0.58 | 0.006 | 0.008 | 0.008 |
| Pb | Soil | 0.80 | 0.69 | 0.78 | 0.011 | 0.018 | 0.015 |
| Pb | Moss | 0.71 | 0.72 | 0.68 | 0.018 | 0.014 | 0.066 |

A special tool for analysing STATISTICA Automated Neural Network (SANN) had been used in training data set and construction of network architectures. This powerful tool has a remarkable ability to detect patterns or trends from quite imprecise data, noisy and complex data but also accurately predict data that are not part of the training data set. Neural networks are highly nonlinear tools that are usually trained using iterative techniques. We have used the BFGS (Broyden-Fletcher-Goldfarb-Shanno) which is one of the two most recommended techniques for training neural networks. In addition to network architectures, the neurons of a network have activation functions that transform the incoming signals from the neurons of the previous layer using a mathematical function. In this data set training, we have used the following set of neuron activation functions for the hidden and output neurons: Identity, Logistic sigmoid, Hyperbolic tangent, Exponential.

Important segment of each advanced machine learning process is to assure simultaneously realistic, comprehensive, and reliable results but in same time the learning process shouldn't be affected with our subjectivity. Thus, it is necessary to establish exact rules for regulating the basic relationship between various data attributes. Otherwise, the subjectivity rapidly increases, if the training data sets is overloaded with the data attributes that support our problem solutions and consequently decrease the autonomy of neural networks.

### 3.2. Model Stability

The main intention of the modelling technique is to prepare stable and repeatable models under the same conditions for both sampling media. It implies that developed approach and procedures can be used without a limitation. For the first time, quite similar approach had been tested at much smaller territory about 100 km$^2$ [17,21] compared to the study area which covers about 5400 km$^2$. The method has been critically evaluated according to the significance of the applied transformations, the similarity between models, as well as the stability of predictions.

Table 3 provides basic statistical parameters of raw vales and predicted values. However, it could be seen, that presented results confirm the previous assumptions. Work with raw values in linear methods does not bring the desired results. Comparing the quartile ranges (P25-P75) and R and R$^2$ values among raw data and predicted values can be concluded that the raw data are showing the highest values. Contrary, the ANN-MP is giving slightly lower values. But anomaly can be explained with the fact that kriging is giving only good results at the point where we collect the sample and depend on the sampling grid (more sampling sites will give better results). Contrary, the ANN is giving much better results even at places that are cover with much lower number of samples, as it will be presented with spatial distribution maps.

**Table 3.** Comparison between the statistical parameters of raw data (n = 409 soil, 286 moss) and predicted values in range of 0.1–99.9 percentiles (n = 539,417).

| Element | Method | Material | X | Md | Min | Max | P$_{25}$ | P$_{75}$ | R | R$^2$ |
|---------|--------|----------|---|----|----|-----|-----|-----|---|----|
| Cu | Row data | Soil | 27 | 26 | 5.8 | 1600 | 19 | 37 | - | - |
| Cu | Kriging | Soil | 24 | 25 | 6.5 | 380 | 19 | 31 | 0.99 | 0.99 |
| Cu | ANN-MP | Soil | 29 | 30 | 14 | 280 | 24 | 36 | 0.73 | 0.54 |
| Pb | Row data | Soil | 37 | 32 | 5.2 | 16,000 | 20 | 80 | - | - |
| Pb | Kriging | Soil | 32 | 29 | 6.6 | 2000 | 21 | 58 | 1.00 | 0.99 |
| Pb | ANN-MP | Soil | 33 | 31 | 7.0 | 6400 | 22 | 56 | 0.78 | 0.61 |
| Cu | Row data | Moss | 6.1 | 5.9 | 1.2 | 300 | 4.6 | 8.4 | - | - |
| Cu | Kriging | Moss | 5.5 | 5.5 | 1.5 | 66 | 4.6 | 6.8 | 1.00 | 1.00 |
| Cu | ANN-MP | Moss | 5.5 | 5.4 | 3.2 | 23 | 4.7 | 6.4 | 0.81 | 0.66 |
| Pb | Row data | Moss | 6.3 | 5.8 | 0.25 | 710 | 3.8 | 11 | - | - |
| Pb | Kriging | Moss | 5.0 | 5.0 | 0.57 | 780 | 3.3 | 7.6 | 1.00 | 1.00 |
| Pb | ANN-MP | Moss | 7.6 | 8.1 | 0.85 | 470 | 4.3 | 15 | 0.72 | 0.52 |

X—mean; Md—median; Min—minimum; Max—maximum; P$_{25}$–P$_{75}$—quartile range, R—correlation coefficient (polynomial regression), R$^2$—coefficient of determination.

The scatterplots with fitted regression lines are showing regressions between transformed values and predictive method (Figure 9). It could be observed that the Box-Cox transformation actually adjusts data to the Gaussian distribution. This could be supported by fact that trace elements are showing a lognormal distribution, but the main elements are normally distributed #. Data transformation is mathematically very complex and specific for each observation, but it gives much better results. From our experience, the operation with the raw data is not recommended at all, since the majority of data is concentrated in area with low values and the regression line is influenced solely by several extreme values [17,21].

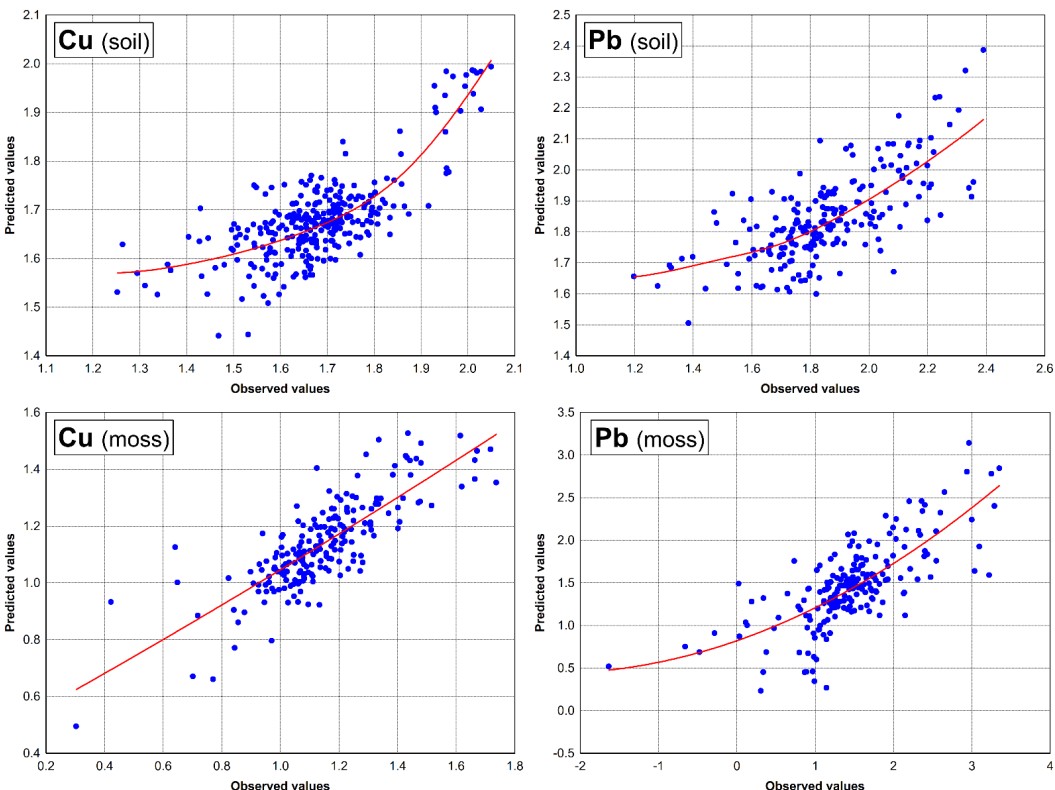

**Figure 9.** Regression plots between predicted and observed values for Cu and Pb in soil and moss (Box-Cox values).

The scatterplots between two applied prediction methods and transformed data are shown in Figure 10. The data cloud represents the real correlations between data obtained using the two different prediction methods. The fitted regression lines were fitting to the majority of predictor values in both sampling media and for both elements, Pb and Cu. It can be observed that the fitted regression lines are sketched using the last squares estimations and fit a set of data points by minimizing the sum of the residual points from the plotted curve. This could be understood as confirmation of our previous claim that methods for its validation must provide similar results.

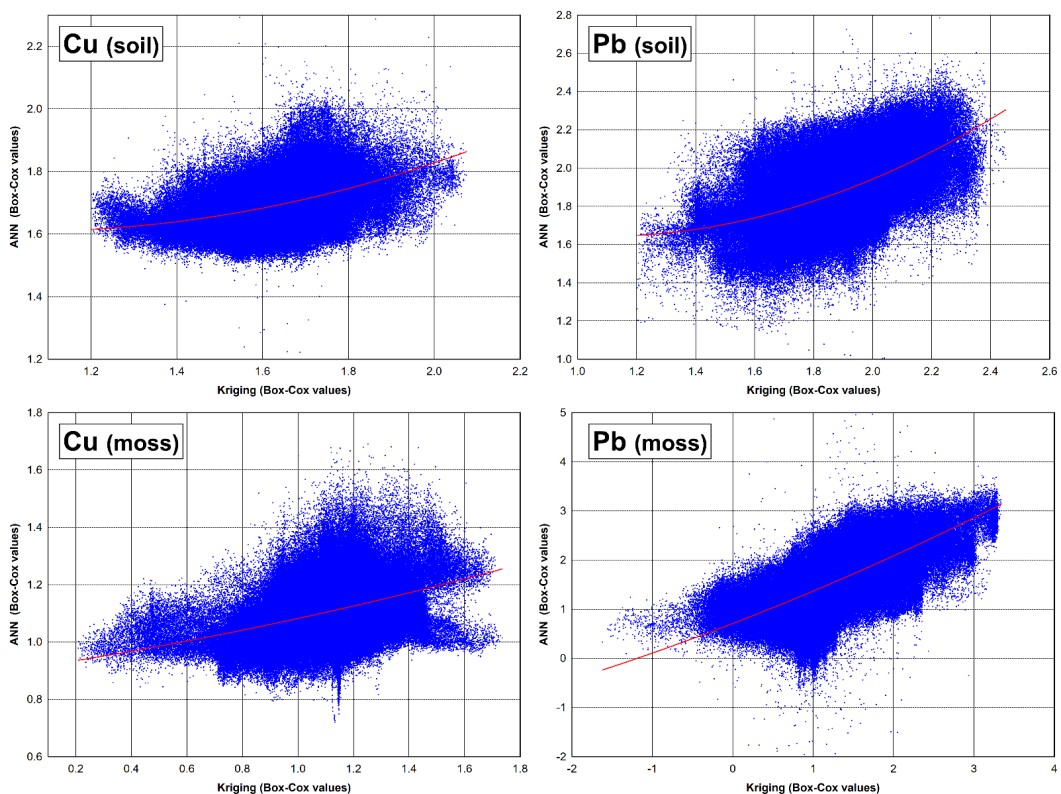

**Figure 10.** Regression plots between prediction models after data transformation (Box-Cox values).

### 3.3. Spatial Distributions

The spatial distributions of lead (Pb) and copper (Cu) in soil and moss are presented in Figures 11–14. For better understanding, the maps of spatial distribution are marked with different colors, each color representing a different degree of contamination. The areas marked in light green represent the lowest level of contamination, while areas marked in red represent the highest level of contamination.

The spatial distributions of Cu in soil and moss produced by the universal kriging model (Figure 11) showed the highest concentrations near the mining areas, obviously. The maps also isolate some areas that might be affected by contamination or appear when areas with the same values around the known data are concentrated (the bullseye effect). In many cases, these enrichments can be enigmatic, incomprehensible, and very demanding in terms of interpretation. Such maps do not provide a clear increase or decrease in the concentrations because they depend only on the analytical measurements and sampling density. Similarly, this was concluded in other studies [51,52]. In order to increase map reliability, in the recent year's authors have dedicated a lots of effort to develop methods that can improve the modelling techniques, and, consequently, the spatial distribution maps as a final product.

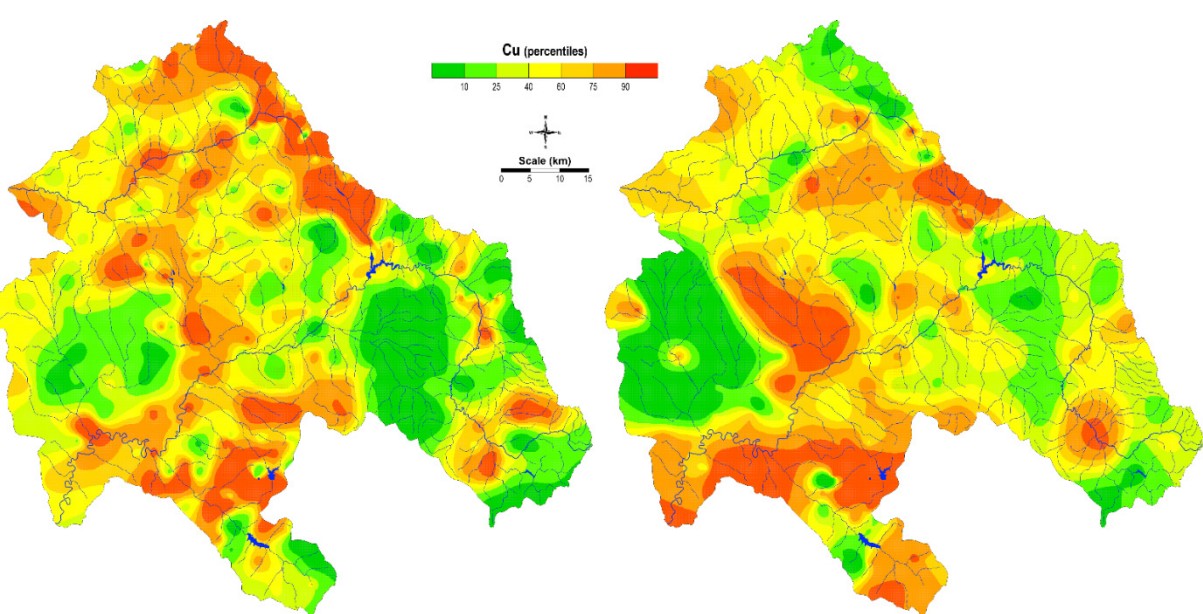

**Figure 11.** Distribution of Cu in soil (**left**) and moss (**right**) after application of universal kriging.

Contrary to this obsolete modelling technique, artificial intelligence has become a favored modeling technique in advanced computer science and gives quite promising results. An obvious improvement can be observed in the distribution maps obtained by ANN-MLP for both sampling materials (Figure 12). The distribution of Cu in soil is very clearly isolated in the area around the Cu mine Bučim, in the southern part of the map, while enrichments in the northern part are related to the lithological units that predominate in the area. Higher levels may also be observed near alluvial plains, which might be a result of both the lithogenic and anthropogenic influences. Specific natural enrichment with copper is found around the Kamenica River and the Zletovska River [53]. Contaminated particles are subsequently transported in water and can be spilt over alluvial deposits during floods, posing a threat to environmental health.

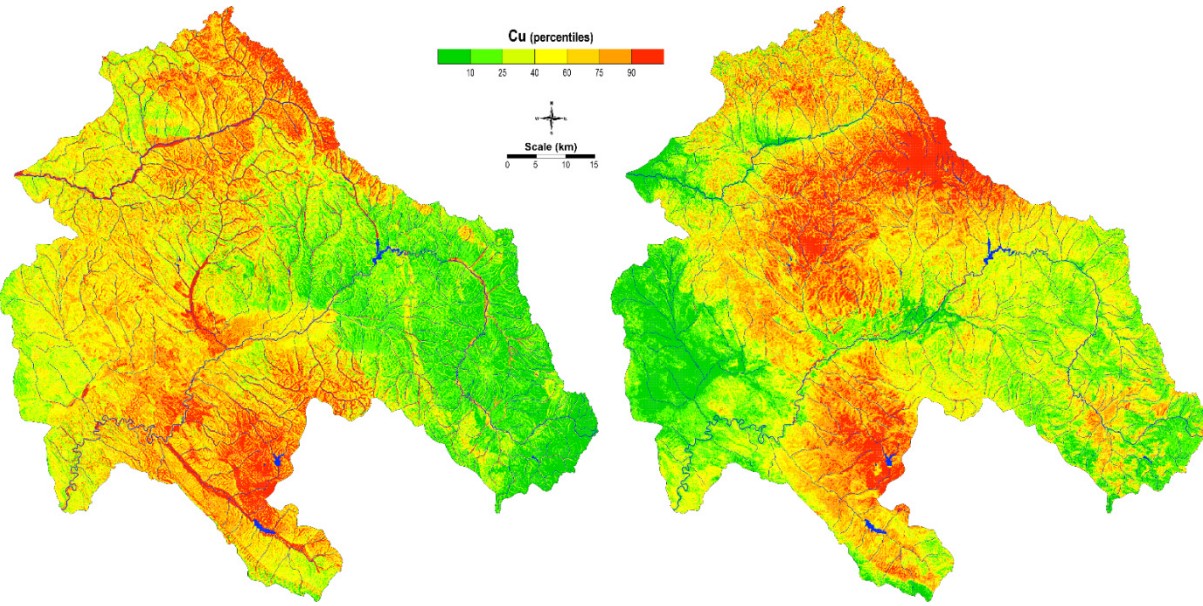

**Figure 12.** Distribution of Cu in soil (**left**) and moss (**right**) after application of ANN-MLP.

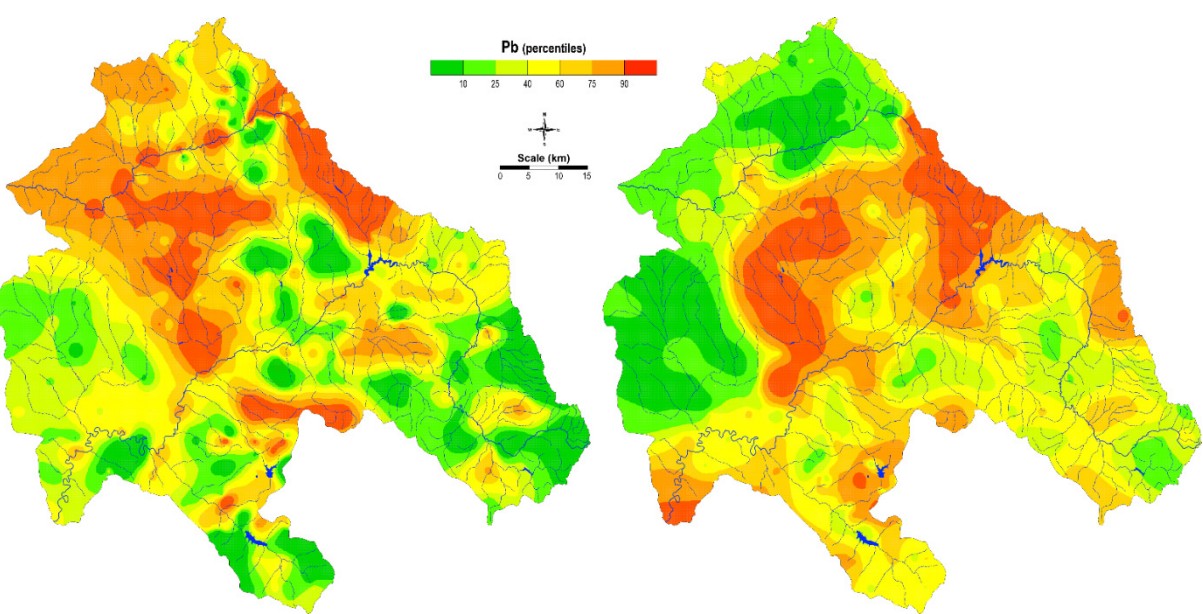

**Figure 13.** Distribution of Pb in soil (**left**) and moss (**right**) after application of universal kriging.

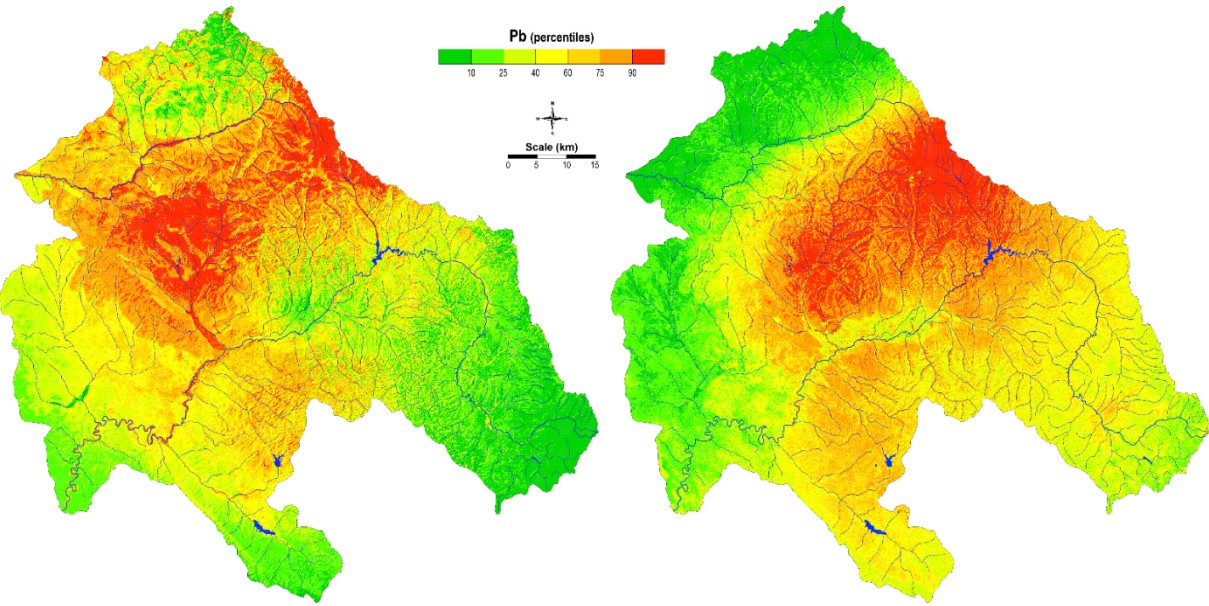

**Figure 14.** Distribution of Pb in soil (**left**) and moss (**right**) after application of ANN-MLP.

The ANN-MLP model for moss, like soil, shows the influence of wind on the spatial distribution of Cu, depending on the main wind direction. The mining of copper minerals at Bučim mine was carried out continuously since 1979, resulting in pollution with airborne dust [54–56]. The highest Cu, as well as Pb, values in mosses, were representative of the surroundings of the Cu Bučim mine, Zn-Pb mines Sasa and Zletovo, while the geological characteristics and lithogenic units did not play a significant role in this case.

The model of the spatial distribution of Pb for soil and mosses using the kriging method gave better results than the previous one (Figure 13). Focusing on Pb enrichment in soil, the spatial distribution was associated with natural enrichment of certain lithological units and was associated Pb mining activity (Zletovo, Sasa-Toranica). However, the enrichment of this element was also detected along the Bregalnica River and its floodplain.

The Pb enrichment in mosses was associated with long-distance transport of polluted air. Along with other elements, such as Cd, Zn, and Sb, Pb enrichment is identified as

an anthropogenic marker for the whole area of North Macedonia [57,58]. There are two dominant sub-areas from which particles are transported by wind: an area of Neogene pyrolusites and volcanic rocks associated with Pb-Zn hydrothermal exploitation ("Zletovo" mine) and an area of Paleogene volcanic sedimentary rocks and Paleogene flysch ("Sasa and Toranica" mine) [28]. Furthermore, during precipitation, these contaminated particles can be also taken up by moss.

Comparing the model using the kriging method with the ANN-MLP method, some similarities can be observed (Figure 14). The enrichment of Pb in soil and mosses was related to Pb mining areas and some lithological units. In addition to wind transport, water transport was also significant, which, in principle, leads to high Pb concentrations in soil. The alluvial plains of the Kamenička and Zletovska Rivers, which flow into surrounding areas, are polluted with lead. Pollution of these areas as a result of mining and mining-related activities has already been pointed out in a previous study [25]. In contrast, high concentrations were not found in the Bregalnica River midstream using ANN. The neural network predicts very precisely the accumulation in Lake Kalimanci, where contaminated sediments settled at the bottom of the lake. Surprisingly, the neural network successfully and precisely predicts data that were not part of our training dataset.

## 4. Conclusions

This works demonstrates the applicability of the ANN multilayer perceptron approach in modeling the spatial distribution of elements. The uniqueness of the study area was emphasized due to its long-term mining and related activities ((i) the Bučim copper mine and flotation, (ii) the Pb-Zn Sasa-Toranica and Zletovo mines, which are a main source of potentially toxic elements), but also because of the complex geological settings, which could be source of natural enrichment and specific geomorphology.

If we compare the two methods, the universal kriging method and the ANN-MLP method, we come to the general conclusion that there are other useful applications besides the ANN-MLP method The spatial distribution of Cu using the kriging method in soils and mosses shows less reliable results due to the so-called "bull's eye" effect, which ignores spatial data such as geological background, pedogenic processes, morphology of the study area, precipitation, wind frequency, and so on. Furthermore, a complex distribution of sampling design cannot provide realistic visualized mapping, especially within a specific topography of the field. The alluvial plains were extracted as the most critical subareas in the research region. The kriging method did not give a visualization of the typical sedimentation effect along the alluvial plains. This is one of the most critical points in favor of using ANN. The ANN-MLP method shows better expression in predicting Cu and Pb distributions in "free space" (not covered areas) and in a cross distance as well. The kriging method may be easier to use, but its results may be erroneous and incoherent because some essential factors have not been considered. Adding various environmental covariates, the constructed maps are providing more reliable results, improve data interpretation and reconstruct main distribution pathways.

The ANN technique provides an unlimited number of algorithms during exploration and consequently provides better and more acceptable results than the universal kriging method, which provides only a limited number of training algorithms. The ANN-MLP method has been improved for demonstrating and predicting the spatial distribution of lead and copper that occurs in a very specific input (anthropogenic, geogenic and even within unpredictable sources).

**Author Contributions:** Conceptualization, R.Š.; Methodology, R.Š. and B.B.; Software, J.A. and R.Š.; Validation, R.Š. and J.A.; Formal analysis, T.S. and B.B.; Investigation, R.Š. and B.B.; Data curation, T.S. and B.B.; Writing—original draft, T.S., B.B., R.Š. and J.A.; Writing—review & editing, T.S., J.A., R.Š., B.B., Visualization, R.Š. and J.A.; All authors reviewed the manuscript. R.Š. and J.A. provided supervision and final approval for the manuscript. All authors have read and agreed to the published version of the manuscript.

**Funding:** This research received no external funding.

**Institutional Review Board Statement:** Not applicable.

**Informed Consent Statement:** Informed consent was obtained from all subjects involved in the study.

**Data Availability Statement:** Data is contained within manuscript.

**Conflicts of Interest:** The authors declare no conflict of interest.

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
