# Peer review of "Multi-Scale Application of Advanced ANN-MLP Model for Increasing the Large-Scale Improvement of Digital Data Visualisation Due to Anomalous Lithogenic and Anthropogenic Elements Distribution"

_minerals, doi:10.3390/min12020174_

Round 1

Reviewer 1 Report

The authors are encouraged to revise the paper according to the following comments:

1) The current literature review section is not comprehensive. More recent works related to natural resource mapping based on GIS, remote sensing, and machine learning should be reviewed and discussed.

2) Why did the authors select ANN in this study? Please explain and provide pertinent references to support your arguments.

3) Why did the authors select kriging as a benchmark model in this study? Please explain and provide pertinent references to support your arguments.

4) Figure 2. Map of soil and mosses sampling site: Some sampling sites are outside the study area. Please double-check this issue?

5) Figure 7: provide legend for different colors in the figure.

6) Section 2.6 and 3.1: describe the input variables and output variable of ANN

7) Section 2.6 and 3.1: describe algorithms for training ANN

8) Section 2.6 and 3.1: How are the number of hidden layers and the number of neurons in these hidden layers selected?

9) How do the authors ensure that severe overfitting has not occurred during the ANN model training phase?

10) Evaluate the model performance in terms of other metrics such as Mean absolute percentage error and coefficient of determination.

Author Response

Answers to the Reviewer 1

No.

Remark

Authors answer

1

The current literature review section is not comprehensive. More recent works related to natural resource mapping based on GIS, remote sensing, and machine learning should be reviewed and discussed.

The remark is accepted, updated reference list Is provided. The review section is added in the introduction part as well in the discussion part as comparative analysis.

The terminology is updated as well.

2

Why did the authors select ANN in this study? Please explain and provide pertinent references to support your arguments.

Comprehensive explanation is given in the introduction part, supported with references listed in the reference list.

3

Why did the authors select kriging as a benchmark model in this study? Please explain and provide pertinent references to support your arguments.

Comprehensive explanation is given in the introduction part, supported with references listed in the reference list.

4

Figure 2. Map of soil and mosses sampling site: Some sampling sites are outside the study area. Please double-check this issue?

Only 7 soils and 5 moss sampling sites are either on the edge or little bit outside from the planned investigated area due to the terrain inaccessibility (canyon area and intensive relief with inaccessible zones). However, they are included in the interpretation since they are containing main information and characteristics of the studied area. Thus, we believe that those several sites should remain as a part of the study area and their given parameters are included in data interpretation.

5

Figure 7: provide legend for different colors in the figure.

The remark is accepted. The authors included a legend in the figure 7.

6

Section 2.6 and 3.1: describe the input variables and output variable of ANN

The remark is accepted. All input data had been added in the manuscript.

7

Section 2.6 and 3.1: describe algorithms for training ANN

Thank you very much for the comment. This part had been revised accordingly in the manuscript.

8

Section 2.6 and 3.1: How are the number of hidden layers and the number of neurons in these hidden layers selected?

Thank you very much for the comment. The complexity of a neural network is measured by the number of neurons in the hidden layer (the more neurons in a neural network, the larger the flexibility and complexity of the system). Based on our experience, the sufficient number of the hidden layers is around 10times higher than number of input variables. In our case we used 250 hidden layers.

9

How do the authors ensure that severe overfitting has not occurred during the ANN model training phase?

Thank you very much for the comment. The SANN tool ensure overfitting automatically and it is based on early stopping.

A separate test set (they never be used in training) is used to halt training to mitigate overfitting. The process of halting of neural network training to prevent overfitting and improving the generalization ability is known as early stopping.

10

Evaluate the model performance in terms of other metrics such as Mean absolute percentage error and coefficient of determination.

Thank you very much for the comment. During the network training data, the random sampling size can be summarized as follows: train 70%, test 15%, and validation 15%. The authors revised this chapter by adding a new table with new parameters.

Reviewer 2 Report

This paper analyzes the spatial distribution of metal pollution (Pb and Cu) throughout two rivers’ basins in a country that is not reported until line 65 (Bosnia and Herzegobina). The work is written for specialists and, in my opinion, should be improved if the authors explain, from the first sentence, which are the aims of the project and, afterwards, which are the methods used; now the order is just the opposite. English is in need of deep revision.

ABSTRACT. The first sentence in the Abstract is not informative: the authors discuss a method without mentioning the main aim of the paper. They introduce “spatial distribution models” of what…

INTRODUCTION. Which is “lithogenic degradation”? It seems that the authors are studying the distribution of two metals, derived from rock sulfides, whereas in the second paragraph they introduce “atmospheric deposition of heavy metals”, which has no relationship with the distribution of metals developed by river networks, the target of this paper. Those ideas are in need of a better introduction. If the distribution is atmospheric AND fluvial, the resulting framework should be better explained.

GEOLOGY. A short description of the volcanogenic rock extension is offered after line 210. However, nothing is said about the hydrothermal activity recorded in the area, probably the most important source of polymetallic sulfides. A summary is introduced in lines 519-525, but nothing is graphically stated.

3.3 SPATIAL DISTRIBUTION. “The spatial distribution of Cu in soil and mosses … are showing the highest concentrations near the mining areas, obviously”. Which is the origin of these mines, volcanic or hydrothermal? The influence of wind is dramatically summarized, as well as that of the hydrographic network, but not quantification is proposed.

CONCLUSIONS. The comparison between both approaches (universal Kriging and ANN-MLP) are compared, though the final result was foreseeable based on the factors considered by each method. I guess readers will be also interested in the relationship between pollution, volcanic/hydrothermal mapping, wind distribution and hydrographic network arrangement. This seems to me an important conclusion that is not considered in its present version.

Author Response

No.

Remark

Authors answer

1

ABSTRACT. The first sentence in the Abstract is not informative: the authors discuss a method without mentioning the main aim of the paper. They introduce “spatial distribution models” of what…

Remark is accepted, more precise information’s are given in the abstract

2

INTRODUCTION. Which is “lithogenic degradation”? It seems that the authors are studying the distribution of two metals, derived from rock sulfides, whereas in the second paragraph they introduce “atmospheric deposition of heavy metals”, which has no relationship with the distribution of metals developed by river networks, the target of this paper. Those ideas are in need of a better introduction. If the distribution is atmospheric AND fluvial, the resulting framework should be better explained.

Remark is accepted, additional information on both approaches has been added to the introduction as well as new references.

3

GEOLOGY. A short description of the volcanogenic rock extension is offered after line 210. However, nothing is said about the hydrothermal activity recorded in the area, probably the most important source of polymetallic sulfides. A summary is introduced in lines 519-525, but nothing is graphically stated.

The remark is accepted. Some information about hydrothermal activity and ore deposits had been added to section Geology.

4

3.3 SPATIAL DISTRIBUTION. “The spatial distribution of Cu in soil and mosses … are showing the highest concentrations near the mining areas, obviously”. Which is the origin of these mines, volcanic or hydrothermal? The influence of wind is dramatically summarized, as well as that of the hydrographic network, but not quantification is proposed.

The quantification approach of the factor impacts and prediction measurements is a total new chapter for investigation. Hopefully we can submit these kinds of data for publication.

5

CONCLUSIONS. The comparison between both approaches (universal Kriging and ANN-MLP) are compared, though the final result was foreseeable based on the factors considered by each method. I guess readers will be also interested in the relationship between pollution, volcanic/hydrothermal mapping, wind distribution and hydrographic network arrangement. This seems to me an important conclusion that is not considered in its present version.

Thank you very much for the comment. This research is based on long-term monitoring in two areas of hydrographic areas. In the preliminary research, the aim was to monitor the distribution of metals and the visualization of the affected zones. Although correction is a very useful tool, we still face a number of anomalies in the interpretation of the results. The application of ANN and the inclusion of additional factor influences has provided great progress in the interpretation and predictability of the distribution of metals whose distribution is strongly anthropogenic.

Therefore, we believe that this preliminary research will lead us to a much more thorough interpretation of the data.

Round 2

Reviewer 1 Report

I have no further comments.

Reviewer 2 Report

The new version has adapted the text to my previous remarks and the final Ms. is now addressed to a broader scientific community. Aims, methods and results are properly documented.